# Flexible TiO_2_/PVDF/g-C_3_N_4_ Nanocomposite with Excellent Light Photocatalytic Performance

**DOI:** 10.3390/polym12010055

**Published:** 2019-12-31

**Authors:** Tong-Tong Zhou, Feng-He Zhao, Yu-Qian Cui, Li-Xiang Chen, Jia-Shu Yan, Xiao-Xiong Wang, Yun-Ze Long

**Affiliations:** 1Collaborative Innovation Center for Nanomaterials & Devices, College of Physics, Qingdao University, Qingdao 266071, Chinawangxiaoxiong69@163.com (X.-X.W.); 2Qingdao SME Public Service Center, Qingdao 266034, China; qdpx@163.com; 3College of Environmental Science & Engineering, Qingdao University, Qingdao 266071, China; bonnie_ie_cyq@163.com; 4Collaborative Innovation Center for Eco-Textiles of Shandong Province, Qingdao University, Qingdao 266071, China

**Keywords:** TiO_2_/PVDF/g-C_3_N_4_, electrospinning, visible light, photocatalyst

## Abstract

As the world faces water shortage and pollution crises, the development of novel visible light photocatalysts to purify water resources is urgently needed. Over the past decades, most of the reported photocatalysts have been in powder or granular forms, creating separation and recycling difficulties. To overcome these challenges, a flexible and recyclable heterostructured TiO_2_/polyvinylidene fluoride/graphitic carbon nitride (TiO_2_/PVDF/g-C_3_N_4_) composite was developed by combining electrospinning, sintering and hydrothermal methods. In the composite, PVDF was used as a support template for removing and separating the photocatalyst from solution. Compared with pure TiO_2_, the TiO_2_/PVDF/g-C_3_N_4_ composite exhibited the extended light capture range of TiO_2_ into the visible light region. The photogenerated carriers were efficiently transferred and separated at the contact interface between TiO_2_ and g-C_3_N_4_ under visible light irradiation, which consequently increased the photocatalytic activity of the photocatalyst. In addition, the flexible composites exhibited excellent self-cleaning properties, and the dye on the photocatalysts was completely degraded by the as-prepared materials. Based on the intrinsic low cost, recyclability, absorption of visible light, facile synthesis, self-cleaning properties and good photocatalytic performances of the composite, the photocatalyst is expected to be used for water treatment.

## 1. Introduction

In recent decades, photocatalysis has been recognized as the most promising and green technologies due to its environmentally friendly, low-cost and high efficiency characteristics. It has been widely used in the fields of environmental management and energy conversion [1,2]. The key of photocatalytic technology is to prepare advanced photocatalysts that can make full use of solar light sources to meet the urgent needs of current environmental treatment. The effective materials commonly used in recent years mainly include semiconductor photocatalysts, such as, metal and non-metal oxides, sulfides and nitrides [3,4]. Among these materials, the good photocatalytic performance and stability of titanium oxide (TiO_2_) have been largely demonstrated. As one of the most outstanding transition metal oxide semiconductor photocatalyst, TiO_2_ exhibits of stable physical-chemical properties, low toxicity and excellent optical performances. Furthermore, it is easily synthesized, and it has a low cost [5,6,7]. However, TiO_2_ is limited in the practical applications, because it has a large band gap energy (3.2 eV) as an n-type wide band gap semiconductor and fast recombination rate of photogenerated electron-hole pairs [8]. These defects lead to low utilization of sunlight and reduce the quantum efficiency, thereby indirectly affecting the photocatalytic performance [9,10]. In addition, TiO_2_ that exists in a powder or particle state is difficult hard to separate from the solution; and even causes secondary pollution to the environment [11,12]. Therefore, it is necessary to find an efficient strategy to simultaneously expand the sunlight capture range and the narrow band gap of TiO_2_, enhance the lifetime of photogenerated electron-hole pairs and facilitate recycling. To date, various methods have been developed, and the design of heterostructured photocatalysts is an effective option [13,14,15]. First, the TiO_2_ compound with a narrow band gap semiconductor can capture the light in the visible region [16]. Second, the matching energy level coupling between the narrow band gap semiconductor and the wide band gap TiO_2_ can promote the transfer and separation of photogenerated carriers, thereby prolonging the lifetime of the photogenerated electron-hole pairs and improving the photocatalytic activity of the photocatalyst [17].

As a narrow band gap semiconductor (2.7 eV), g-C_3_N_4_ exhibits stable physicochemical properties, thermal stability and remarkable optoelectronic transmission performances. Furthermore, it is non-toxic it is easily stored, and it can be obtained from a wide variety of sources [18]. It has been proven in recent years to be an optimal choice for composing heterostructured materials with wide band gap semiconductors. For example, Wang et al. developed a novel in situ microwave-assisted synthesis approach for fabricating N-TiO_2_/g-C_3_N_4_ composites, and the products exhibited remarkable enhancements of their photocatalytic activities [19]. Miranda et al. obtained the g-C_3_N_4_/TiO_2_ composites, which were highly photoactive by combining hydrothermal and sintering methods, and the final conversion rate of phenol was around 90% [20]. Han et al. prepared a g-C_3_N_4_/TiO_2_ composite by integrating electrospinning and calcination technology, and the hydrogen production rate under sunlight was 8931.3 μmol·h^−1^·g^−1^ [21]. However, these prepared materials were still in powder form. Therefore, they were difficult to recycle after the photocatalytic process [22,23]. This caused not only significant waste of the photocatalyst, but also secondary pollution to the environment. At present, a number of methods for recovering inorganic photocatalyst materials have been reported, such as adding magnetic materials, making tough, recyclable and flexible thin film materials, and loading them on organic polymer materials [24,25,26,27,28,29]. Usually, polyvinylidene fluoride (PVDF) is selected as the support material by electrospinning due to its mechanical strength stability, heat resistance and ease of forming fibers [30,31,32]. Our previous work also indicated that the polymer fibers of PVDF prepared by electrospinning could be used as the substrate with inorganic TiO_2_ tightly embedded on the fibers [33].

In this work, flexible TiO_2_/PVDF/g-C_3_N_4_ composites were prepared through electrospinning, sintering and a hydrothermal treatment [34,35]. The hydrothermal method not only successfully coated the PVDF fibers with g-C_3_N_4_, but also in situ formed granular TiO_2_ on the PVDF fibers. The photocatalytic mechanism of the composite during the degradation of pollutants under light has been reasonably explained. The results confirmed that the TiO_2_/PVDF/g-C_3_N_4_ composite extended the response of TiO_2_ to the visible light range and simultaneously significantly improved the photocatalytic activity. In addition, the as-prepared material with the support of PVDF could be recycled easily and contamination by pollutants could be prevented. Therefore, the flexible TiO_2_/PVDF/g-C_3_N_4_ composite is a good candidate for use in the environmental treatment of organic pollutants.

## 2. Experimental

### 2.1. Materials

PVDF (*M*_W_ = 500,000) powder was purchased from Shanghai 3F New Materials Co., Ltd. Melamine (Shanghai, China) (CP, 98.0%) was purchased from Sigma-Aldrich (Shanghai, China). Tetrabutyl orthotitanate (TBOT, CP, 98.0%), rhodamine B (RhB), methyl blue (MB), methyl orange (MO), dimethylformamide (DMF, AR, 99.5%), sulphuric acid (H_2_SO_4_, CP, 95.0%–98.0%) and acetone (CP, 99.0%) were obtained from SinopharmChemical Reagent Co., Ltd (Shanghai, China). Degussa P25 (P25^®^) was purchased from Evonik Degussa Company (Shanghai, China). All chemical reagents were used as received without further purification.

### 2.2. Preparation of TBOT/PVDF Fibers

First 4 g of PVDF powder, 10 mL of DMF and 12 mL of acetone were placed into a 50 mL conical flask and stirred vigorously for 1 h. Next 10 mL of TBOT was added to the solution and stirred continuously for 1 h. The TBOT/PVDF solution was ultrasonically treated for 30 min, and the resultant solution was an electrospinning precursor. TBOT/PVDF fibers were obtained by electrospinning. During the electrospinning process, DC high voltage power was supplied at 10 kV, the pump rate was 2 mL h^−1^, and the tip-to-collector distance was 10 cm. Finally, the collected TBOT/PVDF fibers were dried at 60 °C.

### 2.3. Preparation of g-C_3_N_4_

The g-C_3_N_4_ was prepared following a previously reported calcination method [36]. Typically, 3 g of melamine was uniformly placed in a lidded crucible, after which it was transferred to a muffle furnace and maintained at 550 °C for 4 h to produce the g-C_3_N_4_. The obtained g-C_3_N_4_ was then ground in a mortar for 0.5 h to obtain a yellow powder.

### 2.4. Fabrication of Flexible TiO_2_/PVDF/g-C_3_N_4_ Composite

The prepared TBOT/PVDF fibers were cut into squares with sizes of 2.5 cm × 2.5 cm. Different amounts of g-C_3_N_4_ powder were added into 30 mL of a 0.5 M sulfuric acid water solution and were well stirred for 60 min. The mixture solution was then transferred into a 50 mL polytetrafluoroethylene (PTFE) autoclave containing a TBOT/PVDF square, and underwent hydrothermal treatment at 150 °C for 24 h (as shown in Table 1). The resulting sample was washed with deionized water and anhydrous ethanol for several times. After drying at 60 °C for 10 h, the final product TiO_2_/PVDF/g-C_3_N_4_ (TPCN) was obtained. As a control experiment, TiO_2_/PVDF fibers were also prepared under similar conditions without the addition of g-C_3_N_4_.

### 2.5. Synthesis and Application Process

As shown in Figure 1, TPCN were synthesized by combining the facile electrospinning method, sintering and a controllable hydrothermal process [31]. First, TBOT/PVDF fibers were obtained using convenient and simple electrospinning technology [28,33]. Secondly, g-C_3_N_4_ was collected by one-step sintering [36]. Later, the cut TBOT/PVDF (2.5 cm × 2.5 cm) and a certain amount of g-C_3_N_4_ were placed into a 50 mL reactor for hydrothermal treatment to prepare flexible TPCN. The photocatalyst contaminated with the dyes was subjected to a period of light irradiation to study the photocatalytic performance. Finally, collecting and cleaning the flexible TPCN composite was collected and cleaned from the solution for next reuse.

### 2.6. Characterization

X-ray diffraction (XRD) patterns were collected using a Rigaku SmartLab X-ray diffractometer with Cu-Kα radiation (λ = 1.54178 Å) in the sweep range from 5° to 85° at a scan rate of 5° min^−1^. The morphology and structure images of the samples were obtained using scanning electron microscopy (SEM, JEOL JSM-7800F, JEOL, Tokyo, Japan). X-ray photoelectron spectroscopy (XPS) was performed using by a Thermo Scientific Escalab 250Xi instrument (Thermo Fisher Scientific, Shanghai, China) with an Al Kα X-ray source at room temperature. The photoluminescence (PL) spectrum was obtained using a Hitachi F-4600 fluorescence spectrometer (HITACHI, Tokyo, Japan) to character the recombination behaviors of the photogenerated carriers using an excitation wavelength of 320 nm. The UV-Vis diffuse reflectance spectra (DRS) of the prepared samples were measured using a PERSEE-T9 UV/Vis spectrophotometer with a scanning wavelength range of 200–800 nm with a resolution of 0.2 nm.

### 2.7. Photocatalytic Activity Performance

The photocatalytic activities of the as-prepared samples were evaluated based on the degradation of RhB, a representative water pollutant, under visible light irradiation with an 800 W Xe lamp (equipped with a 420 nm cut-off filter, Beijing Princes Technology Co., Ltd., Beijing, China). The different types of prepared photocatalysts were added into 100 mL quartz tubes containing 50 mL of homogeneous RhB aqueous solution (5 mg L^−1^). The quartz tube was then kept in a dark atmosphere for 30 min while stirring continuously to ensure the photocatalyst surfaces achieved adsorption–desorption equilibrium [37,38]. A Xe lamp was subsequently turned on during the photodegradation process. Circulating water was applied in the reaction system to maintain a constant temperature at 20 °C. Three-milliliter samples of the suspension were removed for centrifugation at specified time intervals. To determine the concentrations of the degraded pollutants, the absorption spectra of RhB in solution after centrifugation were detected using a PERSEE-T9 UV/VIS spectrophotometer (Shanghai Yuan Analysis Instrument Co., Ltd., Shanghai, China). To confirm the recycling stability, the as-prepared composite was washed several times with deionized water followed by drying at 60 °C for 10 h.

## 3. Results and Discussion 

### 3.1. Structure and Morphological Characteristics

The XRD patterns of the as-prepared photocatalysts are depicted in Figure 2. As shown in Figure 2, curve a, only one broad diffraction peak located at 20.4° was detected, which was attributed to the pure β phase of PVDF [39]. This showed that during the electrospinning process, the TBOT did not convert to TiO_2_ with good crystallinity, and only PVDF and the precursors of TiO_2_ were presented in the form of fibers. After undergoing the hydrothermal process, the TBOT began to transform into anatase phase TiO_2_, due to the reactions of the fibers in the 0.5 M H_2_SO_4_ at 150 °C for 24 h, as shown in Figure 2, curve b. The curve contains three sharp 2θ peaks at 25.4°, 48.0° and 54.8°, which were attributed to the (101), (200) and (211) crystal faces of the anatase TiO_2_ (JCPDS 21-1272), respectively [40]. No other diffraction peaks were detected, except for those of the PVDF and anatase TiO_2_. Figure 2, curve f contains one broad diffraction peak at 13.1° and one narrow diffraction peak at 27.5°, which corresponded to the (001) and (002) crystal faces of the g-C_3_N_4_ (JCPDS 87-1526) [41]. In Figure 2, curves c and d, the peaks of TPCN1, TPCN2 and TPCN3 were similar to those of curve b. The peak intensities of PVDF and anatase TiO_2_ began to decrease as the content of g-C_3_N_4_ used in the hydrothermal process increased, indicating that the content of g-C_3_N_4_ grown on the fibers gradually increased and the surface quantities of PVDF and anatase TiO_2_ were masked. Thus, it was concluded that the g-C_3_N_4_ had bonded to the PVDF/TiO_2_ fibers, and the TPCN composites were successfully synthesized using the electrospinning technology and hydrothermal process.

The typical SEM images were used to observe the microscopic features and structures of the as-prepared materials. The initial electrospun TBOT/PVDF fibers were disordered, heterogeneous and interlaced to form a nonwoven material, as shown in Figure 3a. Partial fiber fractures fracture occurred, which mainly occurred because the TBOT in the fibers hydrolyzed due to contact with moisture in the air. Figure 3b reveals that the surface roughness of the TiO_2_/PVDF fibers appeared to be different from the surface roughness of the TBOT/PVDF fibers. Thus, the precursor of TiO_2_ was confirmed to have been successfully doped into the fibers. After the initial product underwent hydrothermal treatment, unevenly distributed particulate matter could be clearly observed on the surfaces of the fibers, which was determined by the XRD patterns to be anatase phase TiO_2_. In the process of acidic hydrothermal growth, the TBOT doped inside and outside the fibers was simultaneously hydrolyzed to TiO_2_. Figure 3c depicts the morphology of the g-C_3_N_4_ prepared using a previously reported method, which possessed a layered structure [42]. As shown in Figure 3d, the fibers were embedded in the g-C_3_N_4_, and the g-C_3_N_4_ was tightly wrapped around the fibers. The XRD patterns in Figure 2 also demonstrated that the TiO_2_ and g-C_3_N_4_ could grow simultaneously on the PVDF fibers by addition of g-C_3_N_4_ in the hydrothermal solution.

The XPS spectra were obtained to determine the surface elemental compositions, chemical bonding states and formation of TiO_2_/g-C_3_N_4_ heterostructures in the photocatalysts. High-resolution elemental analysis was performed to characterize the characteristic peaks of F, C, N, Ti and O. The F 1s spectrum in Appendix A (from Appendix A) shows one major peak centered at 688.2 eV, which was attributed to the presence of the most abundant component, PVDF, in the fibers. [33] In Figure 4a, four carbon species were found in the binding energy region of C 1s spectrum, including 286.5 eV (C–N) and 290.9 eV (F–C–F) [43,44,45]. Meanwhile, the strong peak of N 1s shown in Figure 4b observed at 400.5 eV was related to the sp^2^ hybridized nitrogen (C=N–C) [46]. The results confirmed that the g-C_3_N_4_ was successfully compounded with the fibers. In the Ti 2p profiles (Figure 4c), the spectrum was divided into two peaks centered at 459.4eV and 465.2 eV, corresponding to Ti 2p^3/2^ and Ti 2p^1/2^, respectively, due to the Ti^4+^ of TiO_2_. [47] As displayed in Figure 4d, the O1s peaks at binding energies of 530.7 eV and 532eV were associated with the Ti–O, containing O^2−^, in the TiO_2_ and oxygen in the surface –OH groups, respectively, proving that the TBOT had completely converted to TiO_2_ in the hydrothermal process [48]. Thus, the result was well matched with the results obtained by SEM and XRD.

### 3.2. Optical Characteristics

The PL spectra were used to determine the recombination efficiencies of the photogenerated carriers and characterize the electron hole pair transfer behavior at the contact interface. The PL spectra of the TiO_2_/PVDF fibers and TiO_2_/PVDF fibers with different contents of g-C_3_N_4_ are shown in Figure 5. The TiO_2_/PVDF and P25^®^ both produced four main emission peaks. The emission peak approximately 406 nm was due to the TiO_2_/PVDF, while the peak located at approximately 402 nm was due to the P25^®^. The position of these two peaks may have been caused by the near-band edge emission of TiO_2_ and P25^®^ [49,50]. In the various as-prepared TPCN composites, the emission peaks for TiO_2_/PVDF and P25^®^ were all transferred to a position of 437 nm, which was caused by the combination of g-C_3_N_4_. As the content of g-C_3_N_4_ used in the reaction solution increased, the intensity of the emission peaks for different TPCN contents increased, because g-C_3_N_4_ has a strong recombination rate of photogenerated carriers as previously reported [42]. In addition, five emission peaks still appeared for the TiO_2_/PVDF and P25^®^ between 439 and 495 nm, which were assigned to the oxygen vacancy related defects occurring in the preparation process [51,52].

The UV-Vis absorption spectra in Figure 6 were used to confirm the optical absorption properties of the TiO_2_/PVDF, TPCN1, TPCN2 and TPCN3, which were obtained by converting the UV-Vis diffuse reflectance spectra using the Kubelka−Munk function [53,54]. The curves of the TiO_2_/PVDF suddenly and abruptly decreased at around 400 nm, indicating that there was no response in the visible range, and the response was in the ultraviolet range, owing to the formation of TiO_2_ on the fibers. After adding various contents ofg-C_3_N_4_, the curves exhibited enhanced responses between 400 and 750 nm. The reason for the decrease in absorption in the UV region was that the addition of g-C_3_N_4_ obstructed the ultraviolet absorption of some of the TiO_2_, increasing the absorption in the visible region. Thus, the curves proved that the range of light captured by the photocatalysts extended to the visible light region because the g-C_3_N_4_ was bonded to the TiO_2_/PVDF fibers during the hydrothermal process. We make the tangent intersect the abscissa axis and show the absorbance of various components. As the content of g-C_3_N_4_ added during the hydrothermal process increased, the response of the TiO_2_/PVDF/g-C_3_N_4_ in the visible light region gradually increased. By extrapolating the Kubelka−Munk function, the band gap energy of the as-prepared products was estimated to be 2.6–3.15 eV. As shown in Figure 6, the curves of TPCN1, TPCN2 and TPCN3 exhibited a blue shift in the band gap energy compared to the TiO_2_/PVDF and began to respond to the visible light region.

### 3.3. Photocatalytic and Self-Cleaning Performances

By detecting the change of the RhB concentration under visible light irradiation for a period of time, the photocatalytic performances of the as-prepared composites were evaluated. The sample was treated in a dark environment for 30 min to achieve adsorption equilibrium. Figure 7a shows the RhB degradation curves of the as-prepared photocatalysts under the same conditions, including P25^®^, TiO_2_/PVDF, g-C_3_N_4_, TPCN1, TPCN2 and TPCN3. RhB without an added photocatalyst was used as a blank control group to confirm the inherent stability of the solution under visible light irradiation. As expected, a negligible change in the concentration of RhB occurred. The curves showed that TPCN2 exhibited a higher photocatalytic performance for degrading RhB than other types of photocatalysts. In particular, the photocatalytic activities of various TPCN composites were higher than those of the g-C_3_N_4_ and TiO_2_/PVDF fibers, which indicated that the amount of g-C_3_N_4_ coupled with TiO_2_/PVDF fibers significantly affected the photocatalytic activity during the reaction process. The effective separation of electron and holes pairs and the expanded specific surface area affected the interfacial contact of the g-C_3_N_4_ and TiO_2_ in the composite system. The proper amount of g-C3N4 added to the composite can significantly affected the efficiency of the photocatalysis. The photocatalytic process for the degradation of RhB can also be expressed using pseudo-first-order kinetics, as follows: [55]
–ln(*C/C*_0_) = *k*_app_*t*(1)
where *k*_app_ is the reaction rate constant, and *C*_0_ and *C* are the concentrations of RhB initially and time *t*, respectively. The pseudo-first-order kinetics curves and reaction rate constant of different photocatalysts are shown in Figure 7b and Appendix A. The calculated reaction rate constant of TPCN2 was 0.0083, which were approximately 2.95, 2.41, 2.84, 1.9 and 1.56 times higher than those of TiO_2_/PVDF, P25^®^, g-C_3_N_4_, TPCN1 and TPCN3, respectively. The results indicated that the as-prepared product was suitable as a photocatalyst for eliminating actual environmental pollution. Furthermore, photocatalytic recyclability is an important attribute for engineering applications. Over three successive cycles, Figure 7c shows that the degradation rate of TPCN2 decreased by less than 5%, and similar photocatalytic behavior was maintained.

In Figure 8a–h, the flexible bending performance of the as-prepared sample was demonstrated. By bending at different angles, the fiber membrane still could well been restored. The flexible nature of the sample contributed to the recycling. Under the irradiation of UV light with a wavelength of 254 nm, the self-cleaning properties of the photocatalyst contaminated with MB were measured as shown in Figure 8i,j. Obviously, the color of the photocatalyst contaminated with the dye having a concentration of 10 mg L^−1^ was barely visible when exposed to UV light for 1 h. Similarly, the dye prior polluted by 10 mg L^−1^ RhB was completely removed in Figure 8k,l after irradiation to UV light for 1 h. Veziroglu et al. studied that the residual residue on the CeO_2_-TiO_2_ hybrid structure during oleic acid treatment completely disappeared within 60 min under ultraviolet irradiation [56]; Zhang et al. reported the self-cleaning performance of CuS/PVDF/TiO_2_ was studied by dropping 10 mg L^−1^ of RhB, methyl orange, and methylene blue onto the surface of the composite under visible light. The color of these dyes almost disappeared in about 120 min [57]. As we know, in the photocatalysis process, the photocatalysts are easily attached to dye molecules causing self-pollution, which reduces its photocatalytic performance. The as-prepared products can greatly improve the photocatalytic effect in practical use due to their self-cleaning ability.

### 3.4. Reaction Mechanisms

To further study the possible reaction mechanism of the prepared photocatalyst, a series of comparative experiments of radical scavengers were carried out in the reaction system to determine the active material leading to the photocatalytic process. The scavengers N_2_, EDTA-2Na, AgNO_3_ and TBA were added to the removal of the active substances $O_{2}^{-}$, h, to further study the possible reaction and OH, respectively. As shown in Figure 9a, the photocatalytic properties changed slightly after the addition of EDTA-2Na, AgNO_3_ and TBA, which means that h, e^−^ and ·OH played minor roles in the degradation of RhB. When N_2_ was added to the solution, the degradation efficiency of the CNT2 for RhB decreased sharply, indicating that ∙$O_{2}^{-}$ played a leading role in the photocatalytic degradation of the pollutants. Based on the above results, a photocatalytic mechanism of the CNT degradation of the pollutants under visible light irradiation was proposed. The mechanism of photogenerated carrier transfer and separation under visible light irradiation is shown in Figure 9b. When the light irradiated the sample, an electron (e^−^) was excited from the valence band (VB) of g-C_3_N_4_ to the conduction band (CB), leaving equal numbers of holes (h^+^) in the VB. Therefore, the separation of photogenerated carriers inherent in the material was realized. Meanwhile, an electron (e^−^) of TiO_2_ was excited from the VB to the CB, leaving the same number of holes (h^+^) into VB. Since the VB potential of TiO_2_ was lower than that of g-C_3_N_4_, the electrons were rapidly transferred from the CB of g-C_3_N_4_ to the CB of TiO_2_ through the tightly contact interface. Thus, the holes moved from the VB of TiO_2_ to the VB of g-C_3_N_4_. Eventually, the survival time of the photogenerated carriers was prolonged, which was beneficial to the enhancement of the photocatalytic activity. The electrons transferred to the CB of TiO_2_ were trapped by oxygen molecules in the dye solution, and thus a strong oxidizing oxygen radical (∙$O_{2}^{-}$) formed on the surface of the sample ($O_{2}+e^{-}\to \cdot O_{2}^{-}$) [58]. Meanwhile, hydroxyl radicals (·OH) were also produced on the surface of the photocatalyst due to the holes of the VB transferred to the g-C_3_N_4_ reacting with water molecules in solution, ($H_{2}O+h^{+}\to \cdot \mathrm{OH}$) [59]. The ∙$O_{2}^{-}$ and  ·OH attached to the sample surface had strong oxidizing properties. In the process of contacting with organic pollutant RhB, the RhB finally mineralized into H_2_O and CO_2_ after undergoing the processes of N-demethylation, chromophore cleavage, ring-opening and mineralization [60].

## 4. Conclusions

Heterostructured as-prepared TPCN composites with excellent flexibilities and outstanding visible light photocatalyzed performances and self-cleaning characteristics were developed by combining a facile electrospinning method with a convenient hydrothermal process. The TBOT that unevenly distributed on the surface and inside the fibers converted into granular TiO_2_ during the hydrothermal process, which was tightly embedded on the PVDF fibers. The introduction of the narrow band gap g-C_3_N_4_ in composed the novel TiO_2_/g-C_3_N_4_ heterostructures, ensured that the TPCN composite possessed a lower recombination efficiency of photogenerated carriers and remarkable photocatalytic activity under visible light irradiation. Use of the organic PVDF fibers as the support substrate enabled the composite to achieve self-standing and bendable properties. In contrast to the conventional powder photocatalysts, the TPCN composite could be easily recovered, so the secondary pollution challenge could be overcome. The current work offers a novel synthesis strategy for developing a new-type of recyclable photocatalyst with a visible light response, good photocatalytic performance and self-cleaning characteristics.

## Figures and Tables

**Figure 1 polymers-12-00055-f001:**
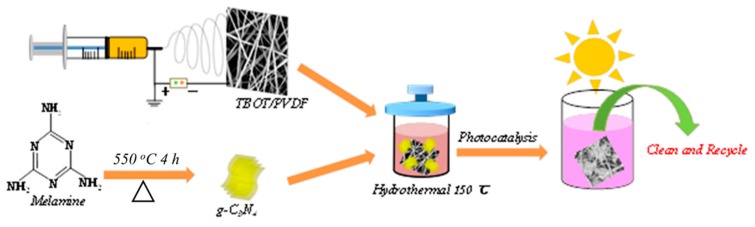
Schematic illustration of synthesis and photocatalytic application of the TPCN composites.

**Figure 2 polymers-12-00055-f002:**
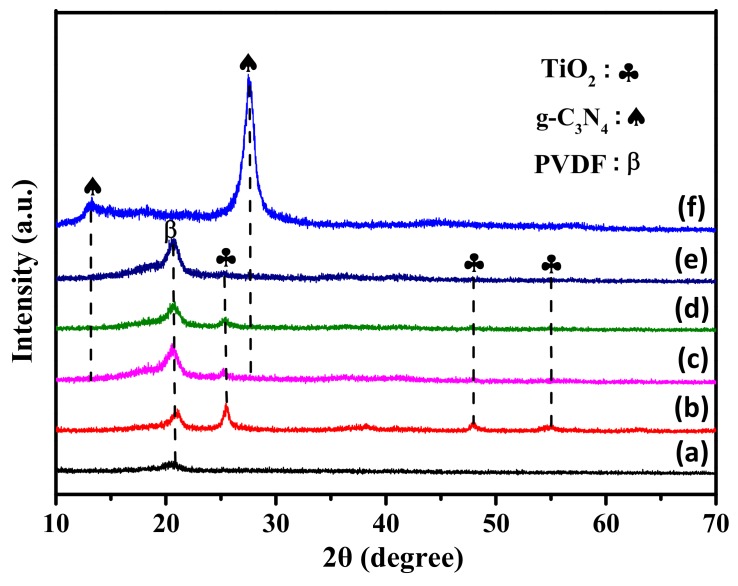
XRD patterns of (**a**) TBOT/PVDF fibers, (**b**) TiO_2_/PVDF fibers, (**c**) TPCN1, (**d**) TPCN2 (**e**) TPCN3 and **(f**) g-C_3_N_4_.

**Figure 3 polymers-12-00055-f003:**
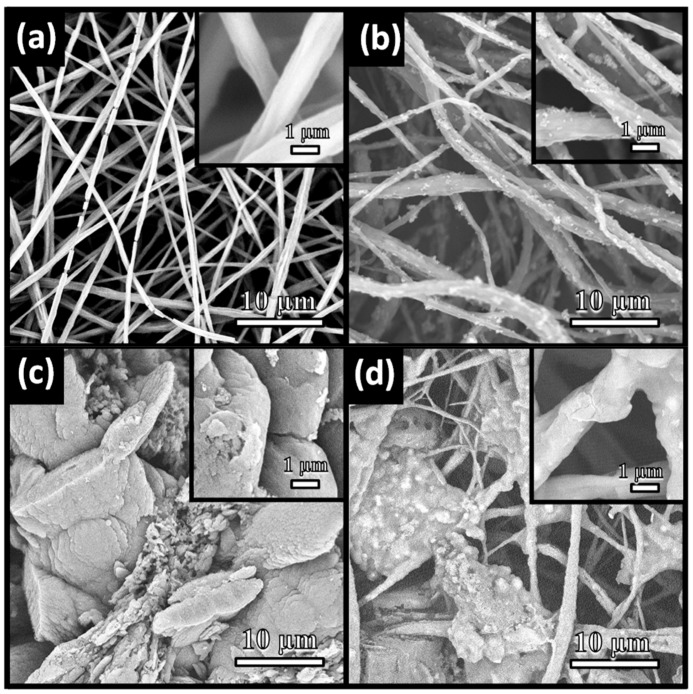
SEM images of (**a**) TBOT/PVDF fibers, (**b**) TiO_2_/PVDF fibers, (**c**) g-C_3_N_4_ and (**d**) TPCN2.

**Figure 4 polymers-12-00055-f004:**
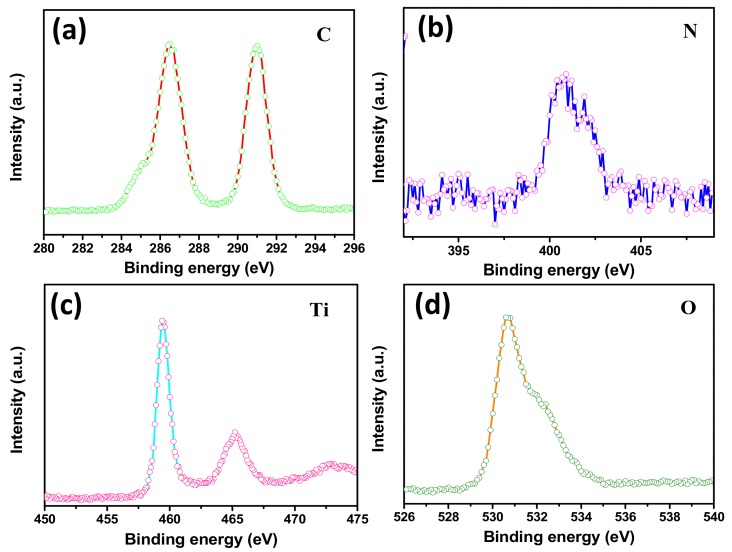
HR-XPS spectra of TPCN2 composite synthesized at 150 °C for 24 h: (**a**) C 1s (**b**) N 1s, (**c**) Ti 2p and (**d**) O 1s.

**Figure 5 polymers-12-00055-f005:**
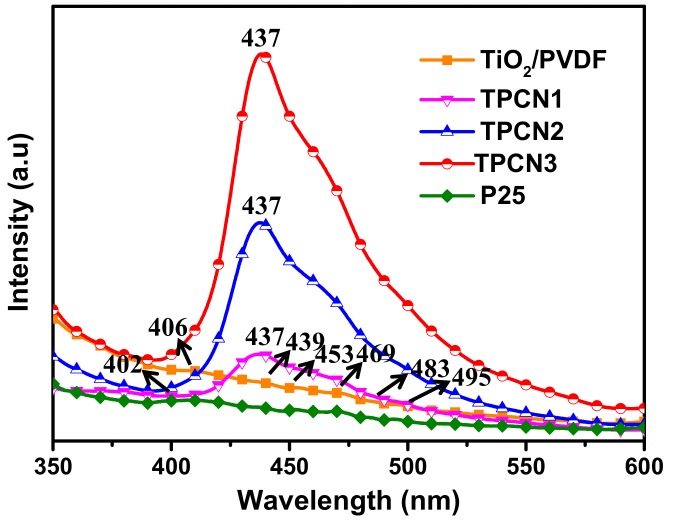
Photoluminescence (PL) spectra of TiO_2_/PVDF, P25^®^ and various as-prepared TPCN composites.

**Figure 6 polymers-12-00055-f006:**
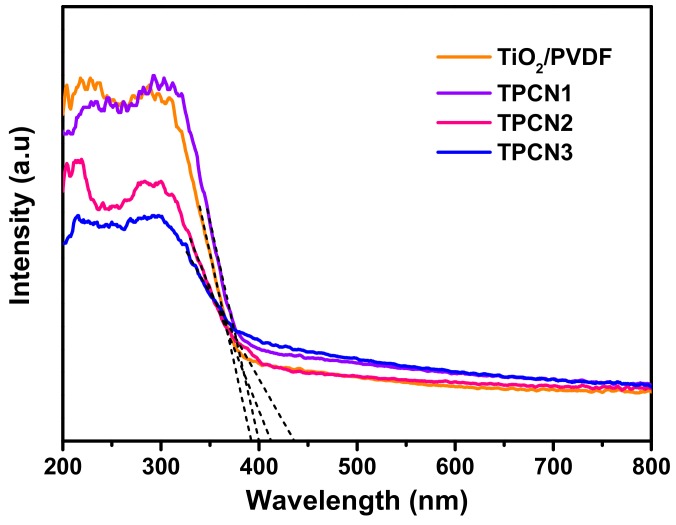
UV-Vis absorption spectra of TiO_2_/PVDF and various as-prepared TPCN composites.

**Figure 7 polymers-12-00055-f007:**
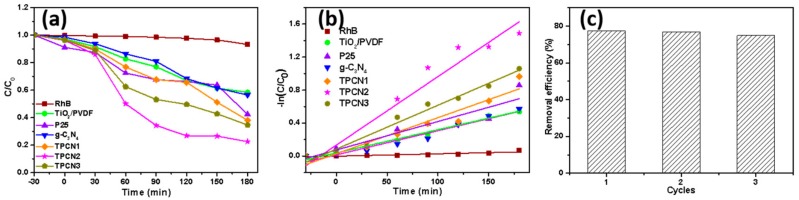
(**a**) Curves of photocatalytic degradation RhB (10 mg/L 50 mL 20 °C) over different samples: RhB bared, P25^®^, TiO_2_/PVDF and as-prepared TPCN composites. (**b**) Kinetic linear fitting curves for photocatalytic degradation of RhB by different photocatalysts. (**c**) The durability of TPCN2 for dye removal after several cycles.

**Figure 8 polymers-12-00055-f008:**
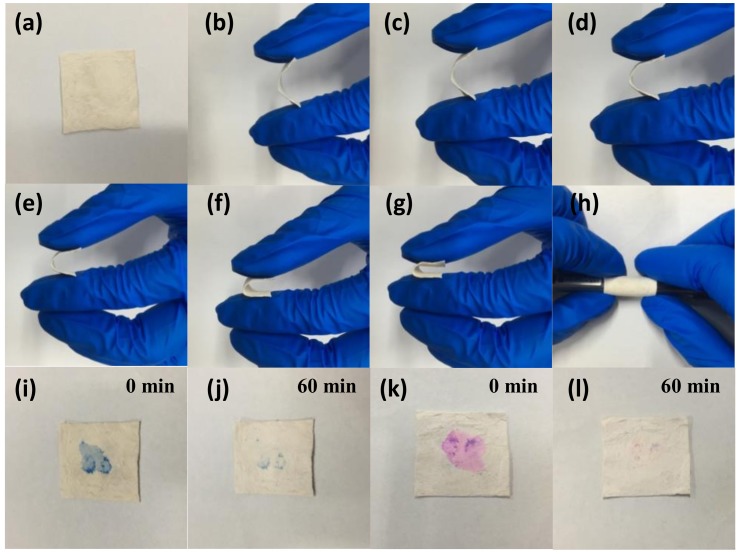
Photographs: flexible performance of TPCN2 (**a**–**h**), stained with MB (**i**,**j**) and RhB (**k**,**l**) under UV light irradiation, respectively.

**Figure 9 polymers-12-00055-f009:**
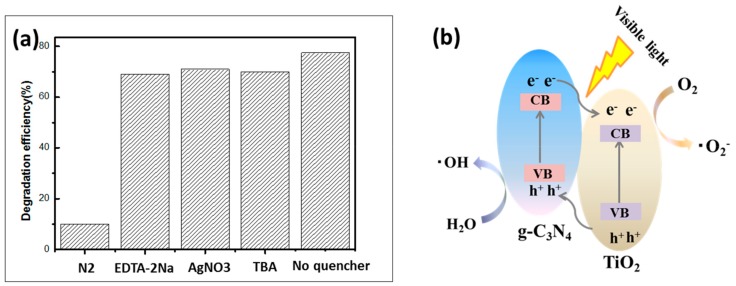
(**a**) Explore the effects of photocatalytic efficiency by adding different types of scavengers. (**b**) Schematic diagrams photocatalytic mechanism of the as-prepared TPCN composite under the visible light irradiation.

**Table 1 polymers-12-00055-t001:** TiO_2_/PVDF/g-C_3_N_4_ (TPCN) samples.

Samples	TPCN1	TPCN2	TPCN3
TiO_2_/PVDF (cm^2^)	2.5 × 2.5	2.5 × 2.5	2.5 × 2.5
g-C_3_N_4_ (mg)	200	600	1000

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
