# Peer review of "Flexible TiO2/PVDF/g-C3N4 Nanocomposite with Excellent Light Photocatalytic Performance"

_polymers, 2019, doi:10.3390/polym12010055_

Round 1
Reviewer 1 Report
This article deals with the interestin idea of making a flexible nanocomposite for water depollution through heterogeneous photocatalysis. The idea is sound, the starting hypothesis good, the experimental design reasonable.
There is a problem with the use of English. The paper has to be proofread by someone proficient in English to correct both the style and the grammar.
The kinetics part of the study is weak, proper comparisons must be established to give a proper idea of the advantages of the fabricated nanocomposite. There are some aspects that have been taken for given, such as the time left for dark adsorption. No information is given about the adsorption-desorption equilibrium. The "reaction mechanisms" is not such, it rather refers to the e-/h+ formation and separation... The reaction mechanism should deal with the reaction products, but these have not been studied... Finally, the authors should report studies on the recyclability / durability of the fabricated material
I regret that I cannot recommend the paper for publication in its current state, but I would be happy to see a revised version of it at a later stage.
Author Response
Reviewer #1
Point 1: The kinetics part of the study is weak, proper comparisons must be established to give a proper idea of the advantages of the fabricated nanocomposite.
Response 1: Thanks. In this work, the authors did comparative experiments on samples with different components. In Section 3.3, the calculated reaction rate constant of TPCN2 was 0.0083, which was approximately 2.95, 2.41, 2.84, 1.9, and 1.56 times higher than those of TiO2/PVDF, P25®, g-C3N4, TPCN1, and TPCN3, respectively.
Point 2: There is a problem with the use of English. The paper has to be proofread by someone proficient in English to correct both the style and the grammar.
Response 2: Thanks. The authors have revised the manuscript carefully, and the English has been polished by native English speaker.
Point 3: No information is given about the adsorption-desorption equilibrium.
Response 3: Dark treatment reached adsorption equilibrium for 30 min, describing in the original page 11, line 305 to 306. The sample was treated in a dark environment for 30 min to achieve adsorption equilibrium.
Point 4: The "reaction mechanisms" is not such, it rather refers to the e-/h+ formation and separation... The reaction mechanism should deal with the reaction products, but these have not been studied.
Response 4: Thanks. The authors have performed a free radical trapping experiment on the corresponding reaction mechanism to better verify the photocatalytic reaction mechanism of the experiment. The scavengers N2, EDTA-2Na, AgNO3 and TBA were added to the removal of the active substances ∙O2-, h, e- and ∙OH, respectively. The photocatalytic properties changed slightly after the addition of EDTA-2Na, AgNO3 and TBA, which means that h, e- and ∙OH played minor roles in the degradation of RhB. When N2 was added to the solution, the degradation efficiency of the CNT2 for RhB decreased sharply, indicating that ∙O2- played a leading role in the photocatalytic degradation of the pollutants. Based on the above results, a photocatalytic mechanism of the CNT degradation of the pollutants under visible light irradiation is proposed.
Point 5: Finally, the authors should report studies on the recyclability/durability of the fabricated material.
Response 5: Thanks. For the recycling and recycling of samples, we added a cycle experiment. It has been proven to be recyclable and reusable. Over three successive cycles, Fig. 7c shows that the degradation rate of TPCN2 decreased by less than 5%, and similar photocatalytic behavior was maintained.

Reviewer 2 Report
I read the manuscript and found it interesting. However, I believe the quality of this paper should further be improved by addressing the following comments.
1- Figures should be appeared after the related paragraphs in the manuscript.
2- Figure 6 does not show a significant absorption in the visible region. It should be modified to see the difference of TiO2/PVDF with other visible-light-active photocatalysts.
3- The used colours in the graphs for each sample should be consistent in Figures 7a and b. Current colours of lines are misleading.
4- Fig 7: It should be explained why the TPCN2 showed lower photocatalytic activity than TPCN3.
5- How many times the developed TiO2/PVDF/g-C3N4 composites can be used? Its recyclability and reusability should be demonstrated.
6- Following references should be added to the text.
i- C. Yao, A. Yuan, H. Zhang, B. Li, J. Liu, F. Xi, X. Dong, Facile surface modification of textiles with photocatalytic carbon nitride nanosheets and the excellent performance for self-cleaning and degradation of gaseous formaldehyde, Journal of Colloid and Interface Science, 533 (2019) 144-153.
ii- E. Pakdel, W.A. Daoud, S. Seyedin, J. Wang, J.M. Razal, L. Sun, et al., Tunable photocatalytic selectivity of TiO2/SiO2 nanocomposites: Effect of silica and isolation approach, Colloids Surf., A. 552 (2018) 130-141.
iii- J. Zhao, J. Wang, L. Fan, E. Pakdel, S. Huang, X. Wang, Immobilization of titanium dioxide on PAN fiber as a recyclable photocatalyst via co-dispersion solvent dip coating, Tex. Res. J. 87 (5) (2017) 570-581.
iv- L. Peng, W. Chen, B. Su, A. Yu, X. Jiang, CsxWO3 nanosheet-coated cotton fabric with multiple functions: UV/NIR shielding and full-spectrum-responsive self-cleaning, Applied Surface Science, 475 (2019) 325-333.

Author Response
Reviewer #2
Point 1: Figures should be appeared after the related paragraphs in the manuscript.
Response 1: Thanks. The figures have been placed after the relevant paragraph of the manuscript.
Point 2: Figure 6 does not show a significant absorption in the visible region. It should be modified to see the difference of TiO2/PVDF with other visible-light-active photocatalysts.
Response 2: Thanks. The authors made the tangent intersect the abscissa axis and showed the absorbance of various components. As the content of g-C3N4 added during the hydrothermal process increased, the response of the TiO2/PVDF/g-C3N4 in the visible light region gradually increased. By extrapolating the Kubelka−Munk function, the band gap energy of the as-prepared products was estimated to be 2.6 - 3.15 eV. As shown in Fig. 6, the curves of TPCN1, TPCN2, and TPCN3 exhibited a blue shift in the band gap energy compared to the TiO2/PVDF.
Point 3: The used colours in the graphs for each sample should be consistent in Figures 7a and b. Current colours of lines are misleading.
Response 3: Thanks. The authors have adjusted the colors used in the graph for each sample in Figures 7a and b. Please see Figure 7.
Point 4: Fig 7: It should be explained why the TPCN2 showed lower photocatalytic activity than TPCN3.
Response 4: Thanks. C0 and C were the concentration of RhB measured at original t0 and temporal t, respectively. The photocatalytic efficiency of TPCN2 was higher than that of TPCN3.
Point 5: How many times the developed TiO2/PVDF/g-C3N4 composites can be used? Its recyclability and reusability should be demonstrated.
Response 5: Thanks. For the recyclability of samples, the authors added a recycling experiment. As shown in Fig. 7c, over three successive cycles, the degradation rate of TPCN2 decreased by less than 5%, and similar photocatalytic behavior was maintained.
Point 6: Following references should be added to the text.
Response 6: Thanks. The documents provided have been added and cited in references 7, 22, 23, and 29.

Reviewer 3 Report
The authors focused on the production of a flexible TiO2/PVDF/g-C3N4 composites through electrospinning. The hydrothermal method was a post-treatment used to coat the PVDF fibres with g-C3N4. The authors proved the visible activity of the produced composite as well as its recyclability. The article presents an innovative material, but there are some issues that should be revised by the authors regarding publication.
(page 2, line 74) In the context of polymeric materials and especially the PVDF and its co-polymers, I recommend the addition of a very recent reference that will enrich state of the art: https://www.mdpi.com/1996-1944/12/10/1649
(page 3, line 98) It is not correct to mention liquids in mass (g) you should refer to volumes (ml).
(page 4, line 146) Figure 1 should be placed on Materials & Methods instead of results and discussion. Additionally, the flame on figure 1 is not adequate for such an illustration.
(page 4, line 147 to 154) The text in section 3.1 is more adequate in M&M… but if you desire to place it on Results and discussion it should be placed after materials characterisation and not before.
(page 4, line 156) You should not start a chapter or subchapter with a Figure.
(page 5) In Figure 3 a) and b) it is strange, the bar scale is the same, but fibres thickness is very different. I would recommend checking the scales of both micrographs, and if they are correct measure fibre diameter using Image J software.
(page 7) In Figure 6, can the authors explain clearly why the nanocomposites material absorbs less in the UV region from TPCN1 to TPCN3?
(page 8, line 242) There is no negative time, change Figure 7a (show a period of time)
(page 8, line 242) The authors should mention the conditions of the assay in subtitles of Figure 7 (temperature, RhB concentration…
(page 8 and 9) there is a lack of comparison of the results with literature, especially regarding the photocatalytic and self-cleaning results. The authors should compare their results with other works and relate such differences with materials properties.
The conclusion lack numbers.
Author Response
Reviewer #3
Point 1: (page 2, line 74) In the context of polymeric materials and especially the PVDF and its co-polymers, I recommend the addition of a very recent reference that will enrich state of the art: https://www.mdpi.com/1996-1944/12/10/1649.
Response 1: The document provided has been added and cited in reference 32.
Point 2: (page 3, line 98) It is not correct to mention liquids in mass (g) you should refer to volumes (ml).
Response 2: Thanks. We convert the solution mentioned from mass (g) to volume (ml). First 4 g of PVDF powder, 10 ml of DMF and 12 ml g of acetone were placed into a 50 ml conical flask and stirred vigorously for 1 h.
Point 3: (page 4, line 146) Figure 1 should be placed on Materials & Methods instead of results and discussion. Additionally, the flame on figure 1 is not adequate for such an illustration.
Response 3: Thanks. We have already placed Figure 1 in the "Materials and Methods" section. In addition, the flame in Figure 1 was modified.
Point 4: (page 4, line 147 to 154) The text in section 3.1 is more adequate in M&M… but if you desire to place it on Results and discussion it should be placed after materials characterisation and not before.
Response 4: Thanks. This part has been adjusted to the experimental part accordingly, and the context is more scientific and coherent.
Point 5: (page 4, line 156) You should not start a chapter or subchapter with a Figure.
Response 5: Thanks. The figures have been placed after the relevant paragraph of the manuscript.
Point 6: (page 5) In Figure 3a) and b) it is strange, the bar scale is the same, but fibers thickness is very different. I would recommend checking the scales of both micrographs, and if they are correct measure fibers diameter using Image J software.
Response 6: Figure 5b shows that the fiber thickness is not uniform. The possible reason is that hydrolysis occurs during the hydrothermal process, causing some of the fibers to become thicker. The fine fibers in Figure 5b are same with Figure 5a.
Point 7: (page 7) In Figure 6, can the authors explain clearly why the nanocomposites material absorbs less in the UV region from TPCN1 to TPCN3?
Response 7: The possible reason for the decrease in absorption in the UV region is that the addition of g-C3N4 covers the UV absorption of part of TiO2, and increases the absorption in the visible region.
Point 8: (page 8, line 242) There is no negative time, change Figure 7a (show a period of time)
Response 8: Here, negative time represents sample absorption time in the dark, describing in the original page 11, line 305 to 306. The sample was treated in a dark environment for 30 min to achieve adsorption equilibrium.
Point 9: (page 8, line 242) The authors should mention the conditions of the assay in subtitles of Figure 7 (temperature, RhB concentration…
Response 9: Thanks. We mentioned in the Section 2.7 that circulating water was applied to the reaction system to maintain a constant temperature of 20 °C. This point has been added in subtitles of Figure 7. Curves of photocatalytic degradation RhB (10 mg/L, 50 ml, 20℃) over different samples: RhB bared, P25®, TiO2/PVDF and as-prepared TPCN composites.
Point 10:- (page 8 and 9) There is a lack of comparison of the results with literature, especially regarding the photocatalytic and self-cleaning results. The authors should compare their results with other works and relate such differences with materials properties.
Response 10: Thanks. The authors have added related materials in page 13 and references 57 and 58. Veziroglu et al. reported that the residual residue on the CeO2-TiO2 hybrid structure during oleic acid treatment completely disappeared within 60 min under ultraviolet irradiation; [57] Zhang et al. studied the self-cleaning performance of CuS/PVDF/TiO2 by dropping 10 mg L-1 of RhB, methyl orange, and methylene blue onto the surface of the composite under visible light. The color of these dyes almost disappeared in about 120 min. [58]
